# Ecological Effects and Microbial Regulatory Mechanisms of Functional Grass Species Assembly in the Restoration of “Heitutan” Degraded Alpine Grasslands

**DOI:** 10.3390/microorganisms13102341

**Published:** 2025-10-11

**Authors:** Zongcheng Cai, Jianjun Shi, Shouquan Fu, Liangyu Lv, Fayi Li, Qingqing Liu, Hairong Zhang, Shancun Bao

**Affiliations:** 1Academy of Animal Husbandry and Veterinary Sciences, Qinghai University, Xining 810016, China; ys230951310630@qhu.edu.cn (Z.C.); ys240951310609@qhu.edu.cn (S.F.); yb230909000074@qhu.edu.cn (L.L.); lfy99218@qhu.edu.com (F.L.); yb220909000082@qhu.edu.cn (Q.L.); 15500620398@163.com (H.Z.); bshancun@163.com (S.B.); 2Key Laboratory of Adaptive Management of Alpine Grassland, Xining 810016, China; 3State Key Laboratory of Ecology and Plateau Agriculture and Animal Husbandry in Sanjiangyuan Jointly Established by the Ministry of Provincial Affairs, Qinghai University, Xining 810016, China

**Keywords:** Qinghai-Tibetan Plateau, grass species mixture, bacterial community structure, plant–soil–microbe interactions

## Abstract

The restoration of “Heitutan” degraded grasslands on the Qinghai-Tibetan Plateau was hindered by suboptimal grass species mixtures, leading to low vegetation productivity, impaired soil nutrient cycling, and microbial functional degradation. Based on a 22-year controlled field experiment, this study systematically elucidated the regulatory mechanisms of different artificial grass mixtures on vegetation community characteristics, soil physicochemical properties, and bacterial community structure and function. The results demonstrated that mixed-sowing treatments significantly improved soil conditions and enhanced aboveground biomass. The HC treatment (*Elymus nutans* Griseb. + *Poa crymophila* Keng ex L. Liu cv. ‘Qinghai’ + *Festuca sinensis* Keng ex S. L. Lu cv. ‘Qinghai’) achieved aboveground biomass of 1580.0 and 1645.0 g·m^−2^, representing 66.14% and 60.91% increases, respectively, compared to the HA monoculture (*E. nutans*). Concurrently, this treatment increased soil organic matter content by 52.3% and 48.4%, total nitrogen by 59.4% and 69.2%, while reducing electrical conductivity by 48.99% and 51.72%, with optimal pH stabilization (7.34–7.38). These findings confirmed that optimized grass mixtures effectively enhance soil physicochemical properties and carbon–nitrogen retention. Microbiome analysis revealed that the HE treatment (*E. nutans* + *P. crymophila* + *F. sinensis* + *Poa poophagorum* Bor. + *Festuca kryloviana* Reverd. cv. ‘Huanhu’) exhibited superior α-diversity indices (OTU, Shannon, Ace, Chao1, Pielou) with increases of 9.36%, 4.20%, 15.0%, 1.76%, and 13.4%, respectively, over HA, accompanied by optimal community evenness (lowest Simpson index). Core bacterial phyla included *Pseudomonadota* (22.7–29.9%), *Acidobacteriota* (21.5–23.6%), and *Actinomycetota* (13.6–16.0%), with significant suppression of pathogenic bacteria. Co-occurrence network analysis identified specialized functional modules, with HC and HD treatments (*E. nutans* + *P. crymophila* + *F. sinensis* + *P. poophagorum*) forming a “nitrogen transformation–antibiotic secretion” network (57.3% positive connections). Structural equation modeling (SEM) revealed that mixed sowing had the strongest direct effect on bacterial diversity (β = 0.76), surpassing indirect effects via soil (β = 0.37) and vegetation (β = 0.11). Redundancy analysis (RDA) identified vegetation cover (24.7% explained variance) and soil pH (20.0%) as key drivers of bacterial community assembly. Principal component analysis (PCA) confirmed HC and HD treatments as the most effective restoration strategies. This study elucidated a tripartite “vegetation–soil–microorganism” restoration mechanism, demonstrating that intermediate-diversity mixtures (3–4 species) optimize ecosystem recovery through niche complementarity, pathogen suppression, and enhanced nutrient cycling. These findings provided a scientific basis for species selection in alpine grassland restoration.

## 1. Introduction

Grassland ecosystems, covering approximately 30% of the Earth’s terrestrial surface, constitute a vital component of global terrestrial ecosystems and play irreplaceable roles in carbon (C) and nitrogen (N) cycling, water and soil conservation, and biodiversity maintenance [1]. China possesses 12.0% of the world’s grassland resources, with the alpine meadows of the Three-River Source Region on the Qinghai-Tibetan Plateau—the core zone of the “Third Pole”—accounting for 50.3% of Qinghai Province’s total area [2]. However, three decades of progressive degradation have transformed over half of these grasslands into “Heitutan”—bare patches resulting from vegetation layer erosion, characterized by 60.0–80.0% reductions in topsoil organic matter (OM) content and vegetation coverage below 15%, leading to near-collapse of ecosystem services [3,4]. The restoration of these degraded grasslands is not only critical for regional ecological security but also provides a key paradigm for rehabilitating fragile high-altitude ecosystems worldwide.

The international research focus in grassland restoration has shifted from singular vegetation recovery to elucidating plant–soil feedback mechanisms [5]. Artificial mixed-seeding techniques have emerged as a preferred strategy for degraded grassland rehabilitation due to their dual benefits of biomass enhancement and soil C sequestration [6]. Based on niche differentiation theory, mixed seeding improves community stability through root spatial partitioning (e.g., deep-and shallow-rooted species combinations) and complementary resource utilization [7]. Saby et al. [8] demonstrated in Mediterranean ecosystems that locally adapted species mixtures significantly increase grassland vegetation density and species richness while promoting soil aggregate formation, thereby enhancing restoration success. However, the unique conditions of alpine ecosystems (mean annual temperature: −4 °C; growing season: 150 days) [9] preclude direct application of temperate-region approaches. Although cold-adapted cultivars such as *Elymus nutans* Griseb. and *Poa crymophila* Keng ex L.Liu cv. ‘Qinghai’ have been developed for the Qinghai-Tibetan Plateau [10], the scientific question of how long-term mixed seeding regulates the tripartite “plant–soil-microorganism” synergistic restoration mechanism in alpine environments remains unresolved.

Soil microbial communities, serving as core drivers of terrestrial ecosystem functions, directly determine nutrient cycling efficiency and system resilience [11]. Among these, bacteria dominate both in diversity and abundance, accounting for 70.0–90.0% of total soil microbial biomass and playing pivotal roles in soil nutrient transformations [12,13]. Current research demonstrates that mixed-seeding combinations exert significant species-specific and functional group-dependent effects on soil bacterial communities [14]. Zhang et al. [15] revealed that mixtures of *Festuca sinensis* Keng ex S. L. Lu and *Poa crymophila* Keng ex L. Liu induced rhizosphere niche differentiation, significantly enriching unclassified *Actinobacteria* and *Nocardioidaceae* compared to monocultures. A global meta-analysis by Chen et al. [16] of 106 paired mixed-versus-monoculture studies confirmed consistently higher microbial diversity in mixed grasslands. Jiang et al. [17] documented in Qinghai-Tibetan alpine grasslands that mixtures of *E. nutans* and *Festuca* spp. increased relative abundances of *Proteobacteria* and *Actinobacteria* by 15.6% and 22.4%, respectively, while decreasing *Acidobacteria*, *Bacteroidetes*, and *Planctomycetes* by 19.3–34.5%. However, existing studies primarily rely on 2–3 year observations, leaving the long-term co-evolutionary mechanisms of plant–soil–microbe interactions during artificial mixed-seeding restoration-particularly the reconstruction of cross-trophic ecological networks systematically unexplored in degraded alpine grasslands.

Building upon this foundation, our study investigates a 22-year artificial mixed-seeding restoration experiment on “Heitutan” degraded grasslands in the Three-River Source Region of the Qinghai-Tibetan Plateau. Through systematic comparison between *E. nutans* monoculture and five mixed-seeding combinations with native grasses—*P. crymophila*, *F. sinensis*, *P. poophagorum*, *F. kryloviana*, and *E. breviaristatus*, this research employs high-throughput sequencing to elucidate the divergence patterns of soil bacterial community structure during long-term vegetation recovery. By integrating vegetation productivity metrics, soil physicochemical properties, and microbial diversity indices, this study specifically aims to:Characterize the composition of key functional bacterial groups and their niche differentiation under varying mixed-seeding ratios;Decipher the multidimensional interaction mechanisms among vegetation traits, soil environment, and microbial networks.

These findings will establish both theoretical foundations and practical solutions for restoring degraded grasslands in alpine ecosystems.

## 2. Materials and Methods

### 2.1. General Situation of the Study Area

The study area was located in Dawu Town, Maqin County, Golog Tibetan Autonomous Prefecture, Qinghai Province, China (100°15′ E, 34°25′ N), situated in the central Qinghai-Tibetan Plateau at an average elevation of 3764 m. The region exhibits a typical alpine frigid climate characterized with a mean annual temperature of −3.9 °C, with January (the coldest month) averaging −12.6 °C, and July (the warmest month) averaging only 9.7 °C, lacking any absolute frost-free period throughout the year. The hydro-thermal regime demonstrates a concentrated precipitation pattern, with an annual mean precipitation of 513.2 mm (predominantly occurring during the monsoon season from June to September) contrasting with a high annual potential evaporation of 2471.6 mm. The pasture growing season lasts approximately 150 days [18].

The study area represents a typical alpine meadow ecosystem dominated by vertical zonal communities comprising Cyperaceae, Poaceae, and Rosaceae species [18]. Soil profiles exhibit characteristic alpine meadow soils with cryoturbation-dominated pedogenesis, showing surface layer (0–20 cm) organic matter content of 8.30–12.6 g·kg^−1^ and pH range of 7.20–8.10 [4]. The pre-existing vegetation had degraded to secondary bare land (“Heitutan”) dominated by noxious weeds including *Aconitum flavum* Hand.-Mazz., *Polygonum sibiricum* Laxm., and *Potentilla anserina* L., with vegetation coverage below 30% and severely impaired ecosystem services, as illustrated in Figure 1c [3,4].

### 2.2. Experimental Materials

The study utilized six *Poaceae* grass species typical of alpine ecosystems: *E. nutans*, *F. sinensis*, *F. kryloviana*, and *E. breviaristatus* (all bunch-type upper-canopy grasses); *P. crymophila* (rhizomatous-bunch type lower-canopy grass); *P. poophagorum* (dense-bunch type lower-canopy grass). All seeds were provided by the Academy of Animal Science and Veterinary Medicine of Qinghai University in Xining, China, with laboratory-confirmed purity ≥ 92.0% and germination potential ≥85.0%.

### 2.3. Experimental Design and Methods

Mixed-Seeding Combinations: Six treatment combinations (HA, HB, HC, HD, HE, and HF) were established using *E. nutans* (the dominant constructive species on the Qinghai-Tibetan Plateau) as the base species, supplemented with five adaptive forage species following functional complementarity principles (Table 1). The mixed-seeding designs adhered to:Vertical stratification pairing of upper-canopy and lower-canopy grasses;Synergistic combination of cold-tolerant and stress-resistant types;Priority selection of high seed-yielding cultivars developed in Qinghai Province.

Plot Establishment and Agronomic Management: The experimental plots were established in 2002 using a randomized complete block design with 18 plots (6 treatments × 3 replicates), each measuring 4 m × 5 m and separated by 2 m buffer zones to minimize edge effects. Prior to sowing, the soil underwent deep loosening tillage (25 cm depth), followed by fragmentation using disc harrows (soil particle size < 2 cm) and final leveling with a land roller. Seeds were drill-sown at 2–3 cm depth with 20 cm row spacing, followed by light roller compaction to conserve soil moisture. Basal fertilizer application consisted of diammonium phosphate (P) (150 kg·hm^−2^) and urea (75 kg·hm^−2^). The seeding rate for each grass species in the mixed-seeding treatments was calculated as its monoculture seeding rate divided by n (where n represents the number of component species in each mixture). Monoculture seeding rates were established as follows: *E. nutans* and *E. breviaristatus* at 3.00 g·m^−2^; *P. pratensis*, *P. crymophila*, and *F. kryloviana* at 0.75 g·m^−2^; and *F. sinensis* at 2.25 g·m^−2^.

Field Management Practices: Prior to plot establishment, systematic rodent eradication was implemented across the experimental area, including both the core zone and a 200-m peripheral buffer zone. Post-establishment monitoring protocols involved annual rodent population surveillance and control measures commencing each March, employing an integrated approach combining bait application and physical barriers to maintain rodent densities below ecological damage thresholds. The grazing management system followed a seasonal regime: complete enclosure protection was enforced during the vegetative growth period (April-November) using 1.5 m high wire mesh fencing, while controlled rotational grazing at strictly regulated stocking rates was permitted during the non-growing season (February–March).

### 2.4. Vegetation Survey and Soil Sampling

Vegetation community parameters including height, coverage, density, and aboveground biomass were investigated annually in August during 2023–2024. Five randomly placed 0.5 m × 0.5 m quadrats were established in each experimental plot. Vegetation height was measured using high-precision steel tapes (±1 mm accuracy) by randomly selecting five individuals per species and calculating the arithmetic mean as plot-level vegetation height.

The plant density (*D*, in plants·m^−2^) was calculated as follows:(1)$$D=\frac{N}{A}$$
where *N* represents total plant counts per quadrat and *A* = 0.25 m^2^.

Canopy coverage (*C*, %) was assessed using a point-intercept method with 10 cm × 10 cm grids (100 intersections per quadrat):(2)$$\mathrm{calculated}\;\mathrm{as}\;C=\frac{n}{100}\times 100\%$$
where *n* = number of vegetation contacts, with overlapping canopies counted only once [19].

Aboveground biomass was harvested at ground level, oven-dried at 105 °C for 30 min (pre-treatment), then at 65 °C to constant weight, and measured using precision balances (0.0001 g accuracy). Detailed vegetation community metrics are presented in Table 2.

Concurrent with vegetation surveys, soil samples were collected using a five-point sampling method with stainless steel augers (5 cm inner diameter) to obtain 0–10 cm surface soil layers. After sieving through 2 mm mesh to remove visible roots and gravels, composite samples from each plot were homogenized and divided into three parts aliquots: (1) air-dried samples stored in dark conditions for analyzing soil organic matter (SOM), total nitrogen (TN), and total phosphorus (TP); (2) samples preserved at −20 °C for determining soil water content (SWC), pH, and electrical conductivity (SEC); (3) samples immediately stored at −80 °C for microbial community structure analysis via soil microbiome sequencing.

### 2.5. Measurements and Methods

Soil physicochemical properties were determined according to the method described by Bao et al. [20]. The following methods were employed: SEC was measured using a portable conductivity meter (Orion Star A211; Thermo Fisher Scientific, Waltham, MA, USA); SOM content was determined by the potassium dichromate oxidation-external heating method; TN was quantified via the Kjeldahl digestion method; TP was analyzed using acid digestion followed by molybdenum-antimony anti-colorimetry; SWC was measured gravimetrically by oven-drying at 105 °C; Soil pH was determined potentiometrically using a pH meter (Mettler Toledo FE28; Mettler-Toledo GmbH, Greifensee, Switzerland) with an electrode system [20].

Soil Bacterial DNA Extraction, PCR Amplification and Sequencing [12,15]: Soil bacterial community structure was characterized using high-throughput sequencing technology, with experimental procedures conducted by Shanghai Majorbio Bio-pharm Technology Co., Ltd. (Shanghai, China) Total microbial genomic DNA was extracted from soil samples using the E.Z.N.A.® soil DNA Kit (Omega Bio-tek, Norcross, GA, USA) according to manufacturer’s instructions. The extraction protocol involved: (1) cell lysis, (2) proteinase K digestion, (3) phenol-chloroform extraction, and (4) ethanol precipitation, ensuring efficient isolation of bacterial genomic DNA. Stringent measures were implemented throughout the process to prevent cross-contamination and maintain DNA integrity.

The extracted DNA was subjected to quality control using a UV spectrophotometer (NanoDrop 2000; Thermo Fisher Scientific, Waltham, MA, USA) to determine nucleic acid purity (target A260/A280 ratio: 1.8–2.0) and concentration (ng/μL), with additional verification of fragment size and integrity performed by 1% agarose gel electrophoresis when necessary to exclude degradation or contamination. Subsequent PCR amplification targeted the V3-V4 hypervariable regions of bacterial 16S rRNA genes using primer pair 338F (5′-ACTCCTACGGGAGGCAGCA-3′) and 806R (5′-GGACTACHVGGGTWTCTAAT-3′). The 25 μL reaction mixture contained: 10 ng template DNA, 0.2 μM of each primer, 1× PCR buffer, 0.2 mM dNTPs, and 1 U Taq DNA polymerase (KAPA HiFi HotStart ReadyMix; Roche Sequencing Solutions, Wilmington, MA, USA). Thermal cycling conditions comprised: initial denaturation (95 °C, 3 min); 30 cycles of denaturation (95 °C, 30 s), annealing (55 °C, 30 s), and extension (72 °C, 30 s); followed by final extension (72 °C, 5 min), with negative controls included to monitor potential contamination. PCR products were separated by 1.5% agarose gel electrophoresis (100 V, 30 min), with the 500 bp target bands excised under UV illumination and purified using commercial kits (Qiagen Gel Extraction Kit; Qiagen, Hilden, Germany) through spin-column adsorption to remove primer dimers and impurities, ensuring final product concentrations ≥20 ng/μL.

Purified PCR products were processed for paired-end sequencing library construction using the Illumina MiSeq PE300 platform. Key steps included: (1) adapter ligation (Nextera XT Index Kit; Illumina, San Diego, CA, USA), (2) size selection (350–600 bp target fragments) via magnetic bead purification, and (3) PCR amplification for library enrichment. Library quality was assessed through concentration measurement (Qubit fluorometer; Thermo Fisher Scientific, Waltham, MA, USA) and fragment distribution analysis (Agilent Bioanalyzer 2100; Agilent Technologies, Santa Clara, CA, USA), with only qualified libraries (≥2 nM concentration, single peak profile) proceeding to sequencing.

High-throughput sequencing was performed on the Illumina MiSeq platform (2 × 300 bp paired-end), with unique barcodes assigned to each sample and a sequencing depth of 50,000 reads/sample to ensure microbial diversity coverage. Raw data were generated in FASTQ format and processed through QIIME2 (v2020.6) pipeline: (1) quality filtering using DADA2 plugin (Q-score < 20, maximum error rate 0.2) to remove low-quality sequences, primer residues, and adapter contaminants; (2) read merging (overlap ≥ 20 bp) to generate non-chimeric sequences; (3) operational taxonomic unit (OTU) clustering at 97% similarity threshold (UCLUST algorithm) with taxonomic annotation against SILVA database (v138, confidence threshold ≥ 80%); and (4) microbial community analysis including α-diversity indices (Shannon, Chao1) and β-diversity metrics (Bray–Curtis dissimilarity) to evaluate structural differences among samples.

### 2.6. Data Analysis and Visualization

The experiment was conducted with three repeated measurements. The raw data were standardized and organized using Microsoft Excel 2019 (Microsoft Corporation, Redmond, WA, USA). Statistical analyses were performed with SPSS 27.0 (IBM Corporation, Armonk, NY, USA), where one-way analysis of variance (ANOVA) was employed to examine differences in vegetation parameters, soil physicochemical properties, and microbial community structure. For significant ANOVA results (*p* < 0.05), Duncan’s multiple range test was conducted for post hoc comparisons with a significance level of α = 0.05. Principal component analysis (PCA) was implemented to extract eigenvalues and eigenvectors of the evaluated parameters, with component scores calculated based on cumulative contribution rates. Composite scores for different mixed-seeding treatments were derived using integrated scoring formulas and subsequently visualized.

Structural equation modeling (SEM) was performed using the lavaan package (v0.6-16) in R (R Foundation for Statistical Computing, Vienna, Austria), Austria. Prior to analysis, all vegetation community indicators, soil physicochemical properties, and bacterial diversity metrics were standardized via Z-score transformation to eliminate unit disparities. Maximum likelihood estimation (ML) was used to calculate path coefficients, with model fit assessed through Fisher’s C test and compared using Akaike’s information criterion (AIC) and Bayesian information criterion (BIC). The standardized path coefficients (β) and their significance levels (*p*-values) quantified effect sizes, while model visualization was achieved using the semPlot package.

Alpha diversity was characterized using five indices: OTU richness (observed OTU counts per sample), Shannon index, Ace index, Chao1 index, and Pielou evenness index. Beta diversity analysis incorporated Bray–Curtis distance-based principal coordinates analysis (PCoA) coupled with PERMANOVA (permutational multivariate analysis of variance) to evaluate bacterial community dissimilarities and inter-group variations. Taxonomic biomarkers were identified through linear discriminant analysis effect size (LEfSe) at both phylum and genus levels (LDA score > 2.0, *p* < 0.05). Functional prediction of soil bacterial communities was performed using FUNGuild via the Majorbio Cloud Platform (https://cloud.majorbio.com (accessed on 15 March 2025)), while redundancy analysis (RDA) elucidated the combined effects of vegetation parameters and soil properties on microbial community structure. Pairwise Pearson correlations between vegetation characteristics, soil physicochemical properties, and microbial diversity metrics were visualized as heatmaps using the OmicStudio online toolkit (https://www.omicstudio.cn/tool/140 (accessed on 22 April 2025)). All graphical outputs were generated with Origin 2022 software.

## 3. Results

### 3.1. Effects of Different Mixed-Seeding Treatments on Soil Physicochemical Properties in Artificial Grassland

As shown in Table 3, different mixed-seeding treatments significantly influenced soil physicochemical properties. In the 21st year after establishment (2023), the HC treatment significantly reduced SEC and pH by 49.0% and 12.0%, respectively (*p* < 0.05), compared with the HA treatment, while simultaneously increasing SOM and TN by 52.3% and 59.4%, respectively (*p* < 0.05). Additionally, SWC and TP reached their highest levels in the HD treatment, exhibiting significant increases of 47.9% and 87.0%, respectively (*p* < 0.05) compared with HA.

As of year 22 (2024), the variation trends in soil characteristics remained consistent with historical observations (Table 3). The HC treatment demonstrated comprehensive improvement, with SWC, SOM, TN, and TP peaking in this treatment—showing increases of 44.6%, 48.4%, 69.1%, and 84.6%, respectively (*p* < 0.05), respectively, compared with HA. Moreover, HC maintained lower soil electrical conductivity (SEC) and pH, with reductions of 51.7% and 10.2%, respectively (*p* < 0.05) relative to HA.

### 3.2. Effects of Different Mixed-Seeding Treatments on Soil Bacterial Community Composition

#### 3.2.1. Quality Assessment of 16S rRNA Sequencing Data and OTU Variation

As illustrated in Figure 2a, the rarefaction curves of all treatment groups exhibited clear plateauing when sequencing depth reached 5000 sequences, indicating that the current sequencing effort sufficiently captured microbial diversity with optimal data saturation. Further sequencing would only detect minimal additional rare taxa. Venn diagram analysis (Figure 2b) revealed that the six mixed-seeding treatments collectively yielded 2206 bacterial operational taxonomic units (OTUs). The HC and HF treatments demonstrated higher species richness, containing 2156 and 2147 OTUs, respectively, whereas HA and HB treatments showed the lowest OTU counts (2088 each). A core microbiome of 1872 OTUs (84.9% of total) was shared across all treatments, while only the HE treatment contained 2 unique OTUs (0.09%), suggesting high bacterial community similarity among treatments with limited treatment-specific taxa.

#### 3.2.2. Taxonomic Composition and Relative Abundance of Soil Bacterial Communities

As shown in Figure 3, the relative abundance of the top 30 dominant bacterial phyla varied significantly among different mixed-seeding treatments. The core dominant phyla, consistently exhibiting the highest relative abundances across all treatments, included *Pseudomonadota* (22.7–29.9%), *Acidobacteriota* (21.5–23.6%), *Actinomycetota* (13.6–16.0%), *Chloroflexota* (5.81–7.68%), *Gemmatimonadota* (4.74–6.63%), and *Bacteroidota* (3.37–5.16%). Notably, microbial community structure displayed marked divergence among treatments. Specifically, compared to the HA treatment, the HE treatment significantly increased the relative abundance of *Actinomycetota*, *Chloroflexota*, and *Gemmatimonadota* while decreasing that of *Pseudomonadota*.

At the genus level (Figure 4), the soil bacterial communities were predominantly composed of norank_o__*Vicinamibacterales* (6.09–7.45%), norank_f__*Gemmatimonadaceae* (4.23–6.30%), norank_o__*Rokubacteriales* (2.53–3.52%), norank_f__*Pyrinomonadaceae* (2.47–3.30%), norank_o__*Subgroup_7* (1.82–3.01%), and *Bradyrhizobium* (1.76–3.16%). Comparative analysis revealed that HB, HC, and HE treatments consistently enhanced the abundance of norank_f__*Gemmatimonadaceae* relative to HA, while HE treatment exhibited reduced levels of norank_o__*Vicinamibacterales* (6.09%) compared to other treatments.

#### 3.2.3. LEfSe Analysis of Soil Bacterial Community Composition

Figure 5 presents the LEfSe results, identifying 31 significantly differentiated biomarkers (LDA score > 2.0) among soil bacterial communities across various mixed-seeding treatments. The analysis revealed substantial heterogeneity in signature microbial taxa between treatment groups, with HA, HD, HE, and HF treatments containing 6, 8, 7, and 10 uniquely responsive taxa, respectively. At the key taxonomic levels: the HA treatment was characterized by dominance of the class *Myxococcia*; the HD treatment showed significant enrichment of the genus *Steroidobacter*; while both HE and HF treatments exhibited distinct differentiation through the families *Aggregatilineaceae* and *Acetobacteraceae*.

### 3.3. Analysis of Soil Bacterial Community Diversity

#### 3.3.1. α-Diversity Analysis of Soil Bacterial Communities

As shown in Figure 6, the mixed-seeding treatments significantly influenced the α-diversity characteristics of soil bacterial communities. The HE treatment exhibited the most favorable microbial diversity profile, with significantly higher values in all measured indices compared to other treatments: observed OTUs (1967), Shannon index (6.43), Ace index (2056), Pielou’s evenness index (0.85), and Chao1 index (2083). These values represented increases of 19.7%, 4.20%, 15.0%, 1.76%, and 13.4%, respectively, relative to the lowest-performing HA treatment. The Simpson index values followed a descending order of HB > HA > HF > HC > HD > HE, with both HD and HE treatments showing significant reductions of 8.50% and 18.0%, respectively, compared to HB. (The detailed computational data can be found in Appendix A).

#### 3.3.2. β-Diversity Analysis (PCoA) of Soil Bacterial Communities

Principal Coordinates Analysis (PCoA) based on Bray–Curtis dissimilarity revealed significant treatment-dependent variations in soil bacterial community structure (*p* = 0.026, Figure 7). The first two principal coordinates (PC1 and PC2) explained 35.5% and 20.7% of the total variance, respectively, with a cumulative contribution of 56.2%. Distinct clustering patterns were observed among treatments: HA, HC, and HF groups exhibited greater β-diversity dispersion, reflecting higher spatial heterogeneity in microbial community composition, while HB, HD, and HE treatments showed more uniform distributions with HE displaying the lowest within-group variation, indicating highly convergent community assembly. Notably, PC2 (20.7% variance explained) effectively discriminated between HA and HE treatments, representing a key driver of community differentiation.

### 3.4. Univariate Correlation Network Variations in Soil Bacterial Communities

Genus-level co-occurrence network analysis (Figure 8) systematically elucidated the impacts of different mixed-seeding treatments on soil bacterial interaction network characteristics. Network topological analysis revealed significant heterogeneity in bacterial network structural parameters across treatments, with edge numbers exhibiting marked variations (392–578) while node counts remained relatively stable. The proportions of positive correlations demonstrated treatment-specific patterns: the HC treatment showed the highest positive edge proportion (57.3%), followed by HF (56.1%), while HA (50.7%) and HD (50.7%) displayed comparable values, and both HB (47.5%) and HE (49.0%) treatments fell below the 50.0% threshold. (The detailed computational data can be found in Appendix A).

Analysis of keystone species revealed significant divergence in core phylum-level connectivity among mixed-seeding treatments. The HA treatment network was characterized by *Verrucomicrobiota*, *Chloroflexota*, and *Actinomycetota* as hub nodes, while the HB treatment exhibited RCP2-54, *Pseudomonadota*, and *Methylomirabilota* as central connectors. Both HC and HD treatments demonstrated convergent dominance of *Pseudomonadota*, *Methylomirabilota*, and *Actinomycetota*, whereas HE and HF treatments shared a common structural framework comprising *Pseudomonadota*, *Chloroflexota*, and *Actinomycetota* as the principal keystone phyla. This treatment-dependent configuration of core microbial assemblages suggested that distinct mixed-seeding regimes potentially regulate soil microbial community assembly through the modification of critical inter-taxa interactions.

### 3.5. Functional Prediction of Soil Bacterial Communities

Functional annotation using FAPROTAX revealed significant treatment-driven differentiation in microbial metabolic profiles (Figure 9). Among the 24 identified metabolic functions, six core functions dominated: chemoheterotrophy, aerobic chemoheterotrophy, animal parasites/symbionts, human pathogens (all), human pathogens (pneumonia), and manganese oxidation. The HD treatment exhibited the highest functional abundance for chemoheterotrophy (5275) and aerobic chemoheterotrophy (5189), followed by HF (4927 and 4848, respectively). In contrast, HA showed prominent representation in animal parasites/symbionts (1905), human pathogens (all) (1563), and human pathogens (pneumonia) (1551), with HC displaying similar functional characteristics (1899, 1513, and 1506, respectively). Notably, HB demonstrated significantly greater manganese oxidation activity (1352) compared to other treatments.

Comparative analysis of functional intensities revealed significant treatment-specific variations (Figure 9). The parasite suppression function (invertebrate_parasites) showed markedly lower intensities in HA (2.33), HC (2.67), HE, and HF (both 2.67) treatments compared to other groups (*p* < 0.05). Notably, HA treatment exhibited the lowest functional intensities for nitrogen_fixation (0.33) and chloroplasts (0.21), while HC treatment demonstrated significantly reduced chloroplasts (1.33) and human_pathogens_nosocomial (2.33) activities.

### 3.6. Cluster Analysis of Soil Bacterial Communities

Hierarchical clustering analysis (Figure 10) revealed significant treatment-driven differentiation in microbial community phylogeny. At a Bray–Curtis similarity threshold of 0.20, all samples segregated into two primary clusters: Cluster 1 comprised solely the HC treatment, while Cluster 2 contained all remaining treatments. Reducing the similarity threshold to 0.15 resolved Cluster 2 into three distinct subclusters: HA and HD treatments formed independent subgroups, while HB, HE, and HF treatments coalesced into a unified subgroup with co-localized soil samples within the same phylogenetic clade, indicating high bacterial community homology.

### 3.7. Heatmap Analysis of Soil Microbial Communities

Phylogenetic heatmap and cluster analysis based on genus-level bacterial composition (Figure 11) demonstrated significant treatment-driven differentiation in soil microbial community structure. The top 50 most abundant bacterial genera exhibited distinct phylogenetic patterns, with HA, HE, HB, and HC treatments showing higher compositional homology, while forming significant divergence from HF and HD treatments. HC treatment displayed a similar genus enrichment profile to HA, characterized by dominant taxa including norank_o__*Vicinamibacterales*, norank_f__*Gemmatimonadaceae*, norank_o__*Subgroup_7*, norank_o__*Rokubacteriales*, and norank_f__*Pyrinomonadaceae*. Notably, both HD and HF treatments exhibited specific selection for *Bradyrhizobium*, with HD additionally enriched in norank_o__*Vicinamibacterales*, norank_f__*Gemmatimonadaceae*, and norank_f__*Pyrinomonadaceae*, while HF contained norank distinguishing feature. HB treatment shared genus enrichment patterns with HE, where HE was primarily dominated by norank_o__*Vicinamibacterales*, norank_f__*Gemmatimonadaceae*, and norank_o__*Rokubacteriales*, whereas HB additionally included norank_o__*Subgroup_7* and norank_f__*Pyrinomonadaceae*.

### 3.8. Coupling Relationships Among Vegetation Characteristics, Soil Physicochemical Properties, and Soil Bacterial Communities

#### 3.8.1. Mantel Test Analysis of Vegetation–Soil–Microbe Interactions

As demonstrated in Figure 12, Mantel test results revealed highly significant positive correlations (*p* < 0.001) among vegetation community parameters (including height, coverage, and density), as well as between these vegetation metrics and aboveground biomass. Concurrently, these vegetation characteristics showed similarly strong positive associations (*p* < 0.001) with key soil physicochemical properties, particularly moisture content, SOM, TN, and TP. In contrast, both SEC and pH exhibited significant negative correlations (*p* < 0.001) with all measured vegetation indicators and other soil parameters.

The bacterial community’s OTU richness exhibited a highly significant correlation with vegetation coverage (*p* < 0.01) and a significant relationship with soil pH (*p* < 0.05), while showing no significant associations with other measured vegetation or soil parameters. Similarly, both the Ace and Chao1 indices demonstrated significant positive correlations with vegetation coverage (*p* < 0.05) but lacked significant relationships with remaining environmental variables. Notably, microbial diversity indices (Shannon, Pielou, and Simpson) showed no statistically significant correlations with any vegetation characteristics or soil physicochemical properties measured in this study.

#### 3.8.2. RDA of Vegetation–Soil–Bacteria Relationships

Redundancy analysis (RDA) was employed to elucidate the relationships between soil bacterial community diversity and environmental factors (vegetation characteristics and soil physicochemical properties) (Figure 13). The first two RDA axes collectively explained 49.8% of the total variation, with RDA1 (39.2%) and RDA2 (10.6%) representing the primary gradients of environmental influence. Vegetation coverage emerged as the most significant driver, accounting for 24.7% of the explained variation in bacterial diversity, followed by soil pH (20.0%). All measured diversity indices (Shannon, Pielou, OTU richness, Ace, and Chao1) showed positive correlations with vegetation parameters (height, coverage, density, and aboveground biomass) and soil nutrient factors (water content, total nitrogen, total phosphorus, and organic matter). Conversely, soil pH and electrical conductivity exhibited consistent negative correlations with all bacterial diversity metrics.

#### 3.8.3. Path Analysis of Vegetation–Soil–Bacteria Relationships

Structural equation modeling (SEM) analysis (Figure 14) demonstrated significant treatment effects on vegetation characteristics, soil properties, and bacterial diversity indices, with excellent model fit statistics (Fisher’s C = 1.12, *p* = 0.57, df = 2, AIC = 23.1, BIC = 32.9). Mixed-seeding treatments exerted a highly significant direct positive effect on bacterial diversity indices (β = 0.76, *p* < 0.001), primarily driving microbial diversity restoration in degraded grasslands. The treatments showed stronger positive effects on soil parameters (β = 0.37, R^2^ = 0.84) than on vegetation indices (β = 0.11, R^2^ = 0.15). Subsequent path analysis revealed that both soil and vegetation indices contributed positively to bacterial diversity (path coefficients = 0.17 and 0.23, respectively), collectively explaining 67.00% of the observed variation.

#### 3.8.4. Comprehensive Evaluation of Vegetation–Soil–Bacteria Systems

Principal component analysis (PCA) (Figure 15a) successfully reduced the dimensionality of 16 vegetation, soil, and bacterial diversity parameters into two orthogonal components, collectively explaining 83.6% of total variance. The first principal component (PC1, eigenvalue = 9.47) accounted for 63.2% of variation, while PC2 explained an additional 20.37%, demonstrating strong systemic correlations among measured indicators.

Treatment-specific PCA scores (Figure 15b) revealed clear differentiation: HC showed the highest positive score on PC1 (1.40), followed by HD (0.98), whereas HE (1.30) and HF (0.59) dominated PC2. Composite PCA scores ranked ecosystem restoration efficacy as HC > HD > HE > HB > HF > HA, indicating that HC most effectively synergized vegetation–soil–microbe rehabilitation, HD and HE demonstrated intermediate performance, and HA yielded the poorest outcomes.

## 4. Discussion

### 4.1. Regulation Effects of Different Mixed-Seeding Treatments on Soil Physicochemical Properties and Vegetation Characteristics in Artificial Grasslands

As fundamental indicators for evaluating grassland degradation status and soil quality, soil nutrients demonstrate significant ecological regulation effects in mixed-seeding artificial grassland systems [21]. Comparative analysis revealed that mixed-seeding restoration significantly enhanced aboveground biomass while simultaneously increasing SOM, SWC, TN content, and TP content compared with monoculture systems, which aligns with previous findings by Zhang et al. [22]. These results confirmed that mixed-seeding practices not only restore grassland productivity by improving soil water absorption and retention capacities, but also promote soil structural stabilization through optimized C and N sequestration mechanisms [23]. The observed significant variations in plant root distribution and biomass among different mixed-seeding treatments were found to intensify spatial heterogeneity of soil nutrient distribution [24]. Particularly noteworthy were the HC treatment and HD treatment, which exhibited the most pronounced improvements in soil physicochemical properties. During the 21st year post-establishment (2023), the HC treatment significantly reduced SEC (49.0%) and pH value (12.0%), while substantially increasing SOM content (52.3%) and TN content (59.4%). These favorable trends persisted into the 22nd year (2024), with further reductions in SEC (51.72%), increases in SOM content (48.4%), and enhanced TN content (69.2%). The HC treatment also achieved the highest vegetation productivity, with aboveground biomass reaching 1580.0 g·m^−2^ and 1645.0 g·m^−2^, representing significant increases of 66.1% and 60.9%, respectively, compared to the HA monoculture treatment. These findings corroborate results from Tong’s [25] study on mixed-grass seeding effects in the Qinghai Lake region, demonstrating that combinations of 3–4 grass species can enhance litter diversity and optimize root exudate composition, thereby facilitating soil C and N accumulation while mitigating salinization stress, ultimately improving forage yield. Supporting evidence from Wang et al. [26] indicates that mixed-seeding artificial grasslands maintain optimal soil pH conditions for vegetation growth, while higher vegetation coverage reduces soil evaporation, preserves soil moisture, and inhibits salt accumulation, thereby synergistically improving both soil water content and pH stability. In the current study, the HC and HD treatments exhibited higher soil water content and pH values within the optimal range for plant growth, suggesting that species-specific root exudate characteristics and salt absorption capacities play crucial regulatory roles in soil pH dynamics, further validating the proposed mechanisms.

In summary, mixed-seeding artificial grasslands effectively reduce interspecific competition intensity through niche differentiation and divergent nutrient utilization strategies among species. Appropriate species combinations can achieve an optimal “complementarity-competition” equilibrium during vegetation restoration in degraded alpine grasslands, while simultaneously enhancing aboveground biomass, increasing soil nutrient contents (SOM, TN, TP), optimizing soil pH value, and improving soil water content. These findings provide critical scientific foundations for quantitative assessment of grassland ecosystem restoration efficacy and dynamic monitoring of plant community health status [27]. Future research should prioritize investigating three-dimensional root system architecture characteristics and interspecific variations in root exudate composition, with particular emphasis on their regulatory mechanisms in soil microenvironments (including C-N transformation and salinization mitigation), thereby further advancing the theoretical framework for sustainable management of artificial grasslands.

### 4.2. Impacts of Different Mixed-Seeding Treatments on Soil Bacterial Community Structure and Function in Artificial Grasslands

Soil microbial diversity serves as a critical indicator for assessing microbial community stability and plays a pivotal role in evaluating soil environmental responses and grassland ecosystem health [28]. Diverse bacterial communities facilitate SOM decomposition and nutrient mineralization, enhance N and P cycling, improve soil aggregate formation and SOM content, reduce water and soil erosion, and maintain biodiversity, thereby supporting the restoration of degraded grasslands [29]. Through systematic analysis of five mixed-seeding treatments and one monoculture treatment, this study confirmed that mixed-seeding treatments significantly optimized bacterial community structure and enhanced both diversity and functionality, consistent with previous research findings [14]. Among all treatments, the HE treatment demonstrated superior performance, exhibiting significantly higher operational taxonomic unit (OTU) richness (1967), Shannon index (6.43), Ace index (2056), Pielou index (0.85), and Chao1 index (2083) compared to the HA treatment (increases of 19.4%, 4.20%, 15.0%, 1.76%, and 13.4%, respectively), while showing the lowest Simpson index value, indicating superior community evenness. These results align with findings reported by Jiang [17] in studies examining the effects of grass mixture seeding on soil microbial diversity in the Qinghai-Tibetan Plateau, suggesting that intermediate-richness mixed-seeding combinations achieve optimal diversity maintenance through niche complementarity, supporting the “intermediate disturbance hypothesis” in ecological theory.

At the phylum level, the study identified *Pseudomonadota* (22.66–29.92%), *Acidobacteriota* (21.5–23.6%), and *Actinomycetota* (13.6–16.0%) as dominant bacterial groups, exhibiting distribution patterns consistent with previous findings by Tao et al. [30]. Mixed-seeding treatments induced significant reductions in soil pH, decreasing from 8.35 in the HA monoculture to 7.34–7.38 in HC mixed-seeding treatments, creating more favorable environmental conditions for acidophilic *Acidobacteriota* proliferation [31]. Furthermore, the HE treatment significantly enhanced the relative abundances of *Actinomycetota*, *Chloroflexi*, and *Gemmatimonadetes*, aligning with research outcomes reported by Kragelund [32] and Nan et al. [33]. These observations suggest the mixed-seeding combination improves soil microenvironments through three primary mechanisms: (1) stimulating proliferation of N-fixing *Actinomycetota*; (2) enhancing nutrient acquisition capabilities of *Chloroflexi*; and (3) promoting plant-growth stimulating functions of *Gemmatimonadetes*. At the genus level, the HE treatment demonstrated distinct regulatory effects on bacterial community structure, simultaneously suppressing norank_o__*Vicinamibacterales* abundance while promoting norank_f__*Gemmatimonadaceae* growth. This phenomenon corresponds with findings from Zhao et al. [14] in Qilian Mountain mixed-grass systems, where interspecific combinations reshaped rhizosphere microecology to suppress potential pathogens while facilitating beneficial bacterial colonization. LEfSe analysis identified 31 differentially abundant biomarkers, with HE treatment-specific enrichment of f__*Acetobacteraceae* potentially serving as keystone functional taxa for soil fertility improvement through nitrogen fixation and plant growth promotion [13].

Co-occurrence network analysis revealed differential regulatory mechanisms of mixed-seeding treatments on soil bacterial interaction networks, demonstrating significant variation in edge numbers (392–578) while maintaining relative stability in node counts (48–50). This observation indicates that mixed-seeding treatments primarily restructure network architecture through modifying microbial interactions rather than increasing species richness, consistent with findings reported by Zhang et al. [15]. The HC treatment exhibited the highest proportion of positive correlations (57.3%), suggesting extensive metabolic complementarity or symbiotic relationships among its bacterial communities, supporting the mutualistic microbial diversity theory proposed by Cai et al. [34]. Contrastingly, the HE treatment showed optimal α-diversity (Shannon = 6.43) but lower positive correlation proportion (49.0%), implying functional redundancy contributes to stability maintenance. Hub analysis demonstrated “core microbiome shifts”: the HA treatment featured *Verrucomicrobiota* as key nodes, known for complex polysaccharide degradation, reflecting monoculture systems’ heavy reliance on C cycling [14]; the HB treatment enriched *Methylomirabilota* (methanotrophs), potentially associated with plant combination-specific methane metabolism promotion [35]; both HC and HD treatments shared *Pseudomonadota-Actinomycetota* hubs, forming a “nitrogen transformation-antibiotic secretion” functional module; while HE and HF treatments exhibited *Chloroflexota* hubization, whose photoheterotrophic characteristics may respond to enhanced litter photodegradation potential in mixed-seeding systems [14,15,36]. These results demonstrate that different plant combinations select specific core functional microbiota through altering resource input quality [35,36], with hub microbe succession driving comprehensive network functional transitions—exemplified by the shift from carbon cycle dominance in HA to nitrogen cycle predominance in HC treatments.

FAPROTAX functional prediction elucidated differential regulatory mechanisms of mixed-seeding treatments on soil bacterial metabolic functions, revealing that the HD treatment exhibited the highest functional abundances for chemoheterotrophy (5275) and aerobic chemoheterotrophy (5189), consistent with previous findings by Zhang et al. [15] and Zhao et al. [14]. Mixed-seeding treatments significantly reduced human pathogens abundance, potentially attributable to antibiotic inhibition by *Actinomycetota* [37]. The HA treatment demonstrated elevated animal parasites functionality (1905), likely associated with ecological niche vacancies resulting from reduced vegetation diversity in monoculture systems [38]. Notably, the HB treatment showed marked enhancement in manganese oxidation functionality (1352), correlated with phenolic compounds (e.g., coumarins) secreted by its specific plant combination (*Elymus nutans Griseb.* + *Poa crymophila* cv. Qinghai) that activate manganese-oxidizing bacteria [15,39]. These findings confirm that mixed-seeding influences ecosystem functionality through a tripartite regulatory network encompassing “resource quality-microbial composition-functional expression.” The HC, HD, and HE treatments demonstrated optimal performance in carbon metabolism, pathogen suppression, and ecological restoration, though further optimization of nitrogen and phosphorus transformation functions remains necessary. These results provide theoretical foundations for targeted restoration of degraded alpine grasslands using grass mixtures, while future research should incorporate metagenomic approaches to elucidate molecular interaction mechanisms among keystone microbial taxa.

### 4.3. Correlation Analysis Between Rhizosphere Microbial Composition and Soil-Vegetation Characteristics

Mantel tests and redundancy analysis (RDA) systematically elucidated the coupling relationships among vegetation, soil, and microorganisms in mixed-seeding artificial grassland ecosystems. The results demonstrated highly significant positive correlations (*p* < 0.001) between vegetation characteristics (height, coverage, and density) and soil nutrient indicators (organic matter, total nitrogen, and total phosphorus), consistent with the “plant–soil feedback” theory proposed by Wang et al. [40]. Significant associations (*p* < 0.05) between bacterial OTU richness, Ace index, Chao1 index and vegetation coverage suggest that canopy cover provides stable microbial habitats by improving microenvironmental conditions and increasing litter input [41]. Soil pH exhibited significant correlations with bacterial diversity metrics, revealing its crucial regulatory role in microbial physiological activities by influencing enzyme activity and membrane stability within optimal ranges [42]. Williams et al. [43] emphasized that pH, available N, P, and SOM content represent core determinants of microbial community composition and diversity. Wardle’s [44] global-scale study identified soil pH and calcium content as primary drivers of bacterial community succession. RDA further quantified vegetation coverage (contribution rate: 24.7%) and soil pH (20.0%) as dominant regulatory variables for bacterial diversity, aligning with Liu et al. [45] findings in Qinghai-Tibetan Plateau grassland restoration. These results demonstrate that plant community structure regulates bacterial diversity through aboveground biomass allocation, which subsequently influences litter input quantity and quality (e.g., C/N ratio), ultimately reshaping soil microbial communities. High vegetation coverage promotes microbial diversity through multiple pathways: increased rhizosphere carbon input, improved soil structure, and microclimate buffering [46], while soil pH shapes bacterial communities by directly modulating cellular metabolism and indirectly affecting nutrient availability [43].

Structural equation modeling (SEM) revealed that mixed-seeding treatments exerted a dominant direct effect (β = 0.76) on microbial communities, significantly surpassing indirect pathways mediated by soil (β = 0.17) and vegetation (β = 0.23). This finding contrasts with traditional “plant–soil–microbe” cascade models [14], suggesting alternative mechanisms: (1) species-specific root exudates directly modulating microbial composition through interspecific plant interactions [15], and (2) litter mixing effects generating novel microbial niches [46]. Principal component analysis (PCA) demonstrated significant treatment-specific impacts on alpine grassland restoration, with PC1 and PC2 collectively explaining 83.60% of variance, confirming strong covariation among vegetation, soil, and microbial indicators—supporting the “vegetation-soil–microbe” synergy hypothesis [47]. The HC treatment achieved the highest PC1 loading (1.40) and composite score (0.71), consistent with its superior soil amelioration and microbial functional optimization, indicating optimal restoration through enhanced nutrient cycling and microbial activation. Notably, the HE treatment (5-species mixture) showed peak PC2 performance (1.30) but ranked third in composite efficacy, suggesting that excessive species richness (5–6 species) may induce functional redundancy and diminish restoration efficiency [48], aligning with the intermediate disturbance hypothesis that predicts peak ecosystem functionality at moderate complexity levels [17]. The HA monoculture yielded the lowest composite score, confirming limitations of single-species systems in maintaining multifunctionality [49]. These findings underscore the critical role of optimized species combinations in alpine grassland restoration, providing theoretical foundations for rehabilitating “Heitutan” degraded grasslands. Future investigations should employ metagenomics to elucidate environment-functional gene interactions at molecular resolution.

## 5. Conclusions

This study systematically elucidates the synergistic regulatory mechanisms and practical implications of plant–soil–microbe interactions in degraded alpine grassland ecosystems. The findings demonstrate that intermediate-diversity mixtures (3–4 species), particularly the combination of *E. nutans*, *P. crymophila*, and *F. sinensis* (HC treatment), as well as the four-species mixture incorporating *P. poophagorum* (HD treatment), significantly enhance ecosystem functionality through niche complementarity. These configurations not only mitigate interspecific competition but also synergistically improve soil physicochemical properties, including reduced salinity-alkalinity stress, enhanced organic matter accumulation, and increased nitrogen transformation efficiency.

Crucially, this study reveals that mixed planting systems drive functional optimization by directionally modulating microbial community structure, establishing distinct functional modules. The 3–4 species mixtures form a “nitrogen transformation-antibiotic secretion” functional hub, while higher-diversity mixtures induce a photoheterotrophic metabolic network. These findings provide critical theoretical and practical insights for grassland restoration.

In practical applications, the intermediate-diversity strategy offers notable advantages: (1) simple and feasible species combinations suitable for large-scale implementation in plateau regions; (2) rapid functional recovery via microbial activation; and (3) a balanced approach ensuring both short-term rehabilitation and long-term stability. These results not only present an optimized solution for alpine grassland restoration but also offer transferable ecological principles for rehabilitating other degraded ecosystems.

## Figures and Tables

**Figure 1 microorganisms-13-02341-f001:**
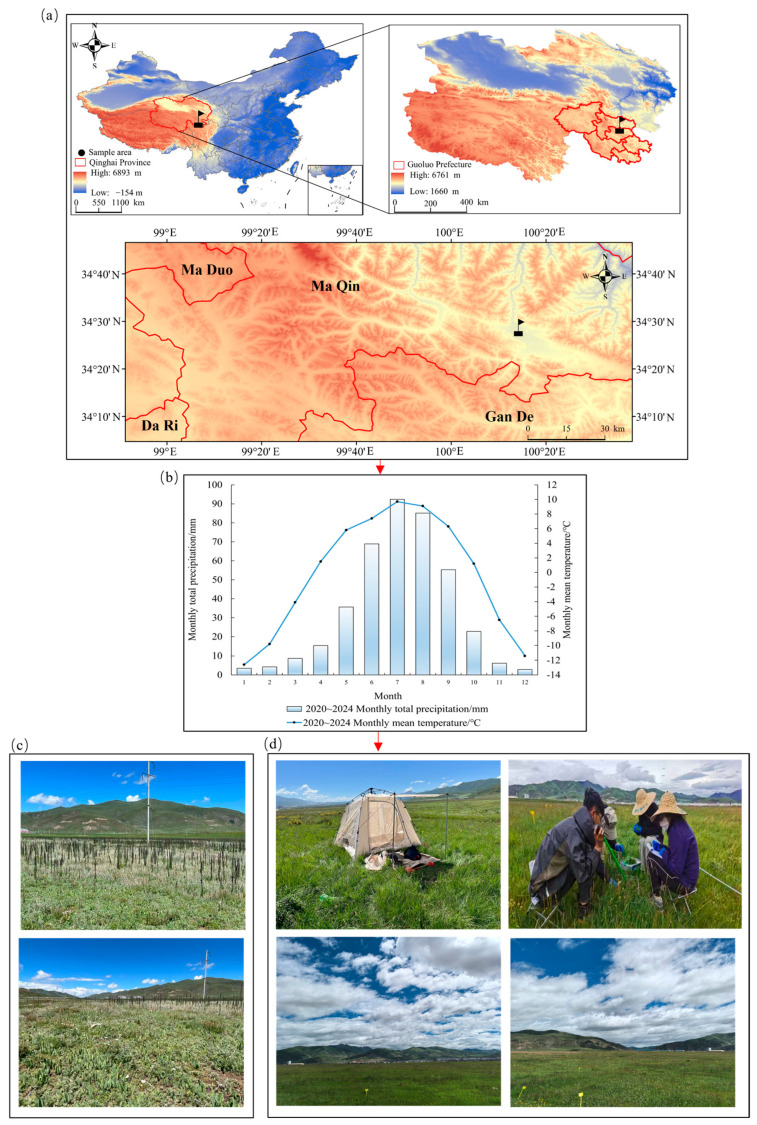
Study area, experimental site, and sampling design. (**a**) Geographical location of the experimental site; (**b**) Monthly mean temperature and precipitation at the experimental site from 2020 to 2024 (meteorological data obtained from the National Meteorological Information Center of China Meteorological Administration, with supplementary regional observational data for Maqin County referenced from publications such as Research on Climate Change in the Qinghai Plateau); (**c**) Appearance of unrestored “Heitutan” degraded grassland; (**d**) Vegetation status of “Heitutan” degraded grassland after 22 years of restoration using artificially established mixed-seeding grassland.

**Figure 2 microorganisms-13-02341-f002:**
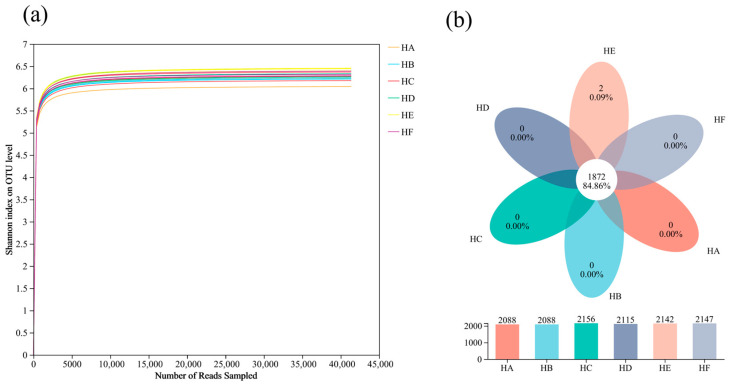
Soil sample rarefaction curves and Venn diagram analysis. Panel (**a**) displays rarefaction curves, and panel (**b**) presents the Venn diagram of OTU distribution. Note: HA (monoculture of *E. nutans*), HB (binary mixture of *E. nutans* + *P. crymophila*), HC (ternary mixture of *E. nutans* + *P. crymophila* + *F. sinensis*), HD (four-species mixture of *E. nutans* + *P. crymophila* + *F. sinensis* + *P. poophagorum*), HE (five-species mixture of *E. nutans* + *P. crymophila* + *F. sinensis* + *P. poophagorum* + *F. kryloviana*), and HF (six-species mixture of *E. nutans* + *P. crymophila* + *F. sinensis* + *P. poophagorum* + *F. kryloviana* + *E. breviaristatus*). The numerical values represent the total number of OTUs in each group, with the overlapping area indicating the quantity and proportion of shared OTUs. The total OTU count was calculated by dividing the number of shared OTUs by their corresponding proportion.

**Figure 3 microorganisms-13-02341-f003:**
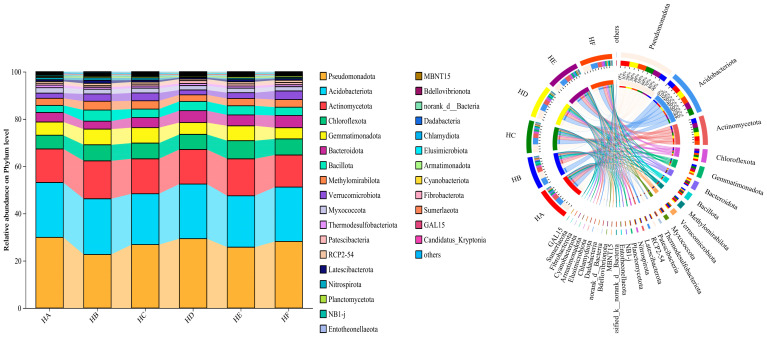
Phylum-level relative abundance of soil bacterial communities across different mixed-seeding treatments.

**Figure 4 microorganisms-13-02341-f004:**
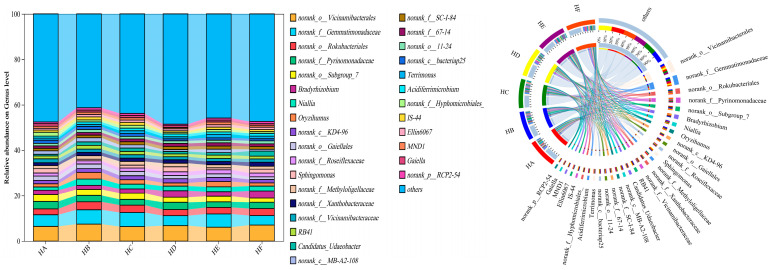
Genus-level composition variations in soil bacterial communities among experimental treatments.

**Figure 5 microorganisms-13-02341-f005:**
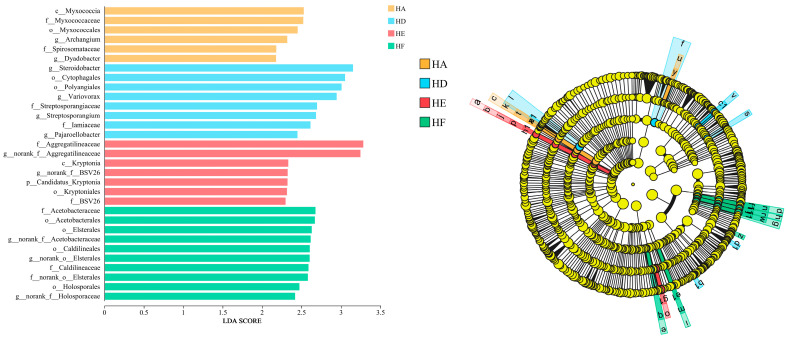
Illustrates the LEfSe analysis of soil bacterial communities under different mixed-seeding regimes.

**Figure 6 microorganisms-13-02341-f006:**
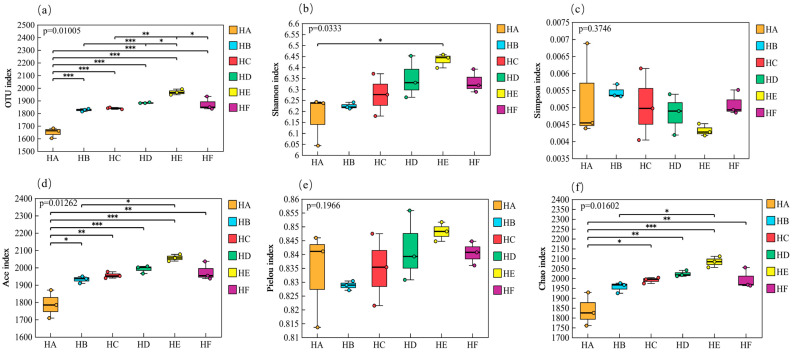
Variations in α-diversity indices of soil bacterial communities across different mixed-seeding treatments. Note: HA (monoculture of *E. nutans*), HB (binary mixture of *E. nutans* + *P. crymophila*), HC (ternary mixture of *E. nutans* + *P. crymophila* + *F. sinensis*), HD (four-species mixture of *E. nutans* + *P. crymophila* + *F. sinensis* + *P. poophagorum*), HE (five-species mixture of *E. nutans* + *P. crymophila* + *F. sinensis* + *P. poophagorum* + *F. kryloviana*), and HF (six-species mixture of *E. nutans* + *P. crymophila* + *F. sinensis* + *P. poophagorum* + *F. kryloviana* + *E. breviaristatus*). (**a**) Number of OUTs, (**b**) Shannon index, (**c**) Simpson index, (**d**) Ace index, (**e**) Pielou index, (**f**) Chao1 index. Data are presented as mean ± standard deviation (*n* = 3). Non-significant differences (*p* ≥ 0.05) are indicated by “ns”. Statistical significance is denoted as follows: * *p* < 0.05, ** *p* < 0.01, and *** *p* < 0.001.

**Figure 7 microorganisms-13-02341-f007:**
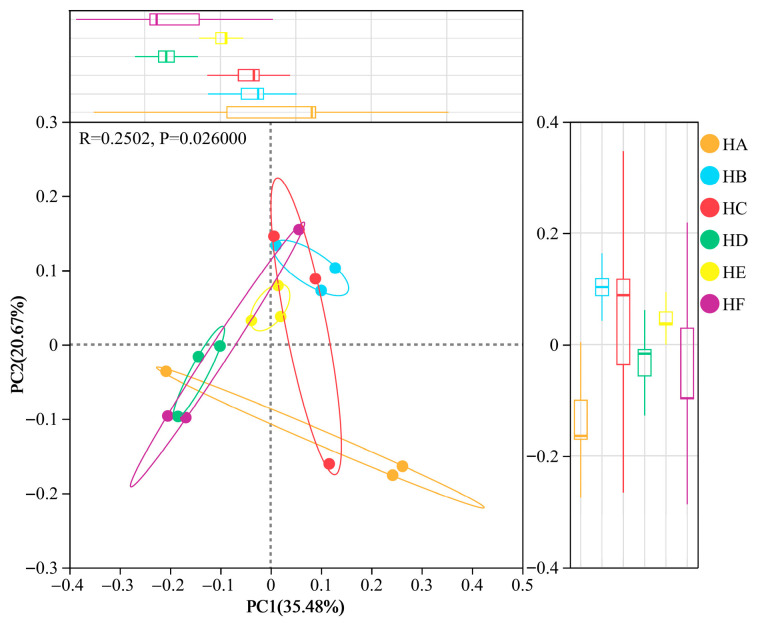
β-Diversity variations in soil bacterial communities across different mixed-seeding treatments based on PCoA analysis (Bray–Curtis dissimilarity). Note: HA (monoculture of *E. nutans*), HB (binary mixture of *E. nutans* + *P. crymophila*), HC (ternary mixture of *E. nutans* + *P. crymophila* + *F. sinensis*), HD (four-species mixture of *E. nutans* + *P. crymophila* + *F. sinensis* + *P. poophagorum*), HE (five-species mixture of *E. nutans* + *P. crymophila* + *F. sinensis* + *P. poophagorum* + *F. kryloviana*), and HF (six-species mixture of *E. nutans* + *P. crymophila* + *F. sinensis* + *P. poophagorum* + *F. kryloviana* + *E. breviaristatus*).

**Figure 8 microorganisms-13-02341-f008:**
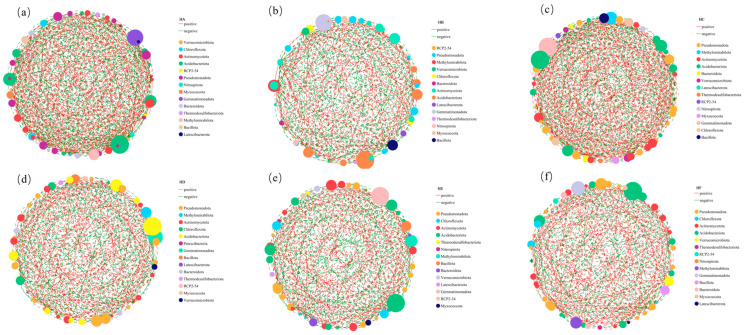
Univariate correlation networks of soil bacterial communities across different mixed-seeding treatments. Note: (**a**) Single-factor correlation network diagram for HA treatment; (**b**) Single-factor correlation network diagram for HB treatment; (**c**) Single-factor correlation network diagram for HC treatment; (**d**) Single-factor correlation network diagram for HD treatment; (**e**) Single-factor correlation network diagram for HE treatment; (**f**) Single-factor correlation network diagram for HF treatment. HA (monoculture of *E. nutans*), HB (binary mixture of *E. nutans* + *P. crymophila*), HC (ternary mixture of *E. nutans* + *P. crymophila* + *F. sinensis*), HD (four-species mixture of *E. nutans* + *P. crymophila* + *F. sinensis* + *P. poophagorum*), HE (five-species mixture of *E. nutans* + *P. crymophila* + *F. sinensis* + *P. poophagorum* + *F. kryloviana*), and HF (six-species mixture of *E. nutans* + *P. crymophila* + *F. sinensis* + *P. poophagorum* + *F. kryloviana* + *E. breviaristatus*).

**Figure 9 microorganisms-13-02341-f009:**
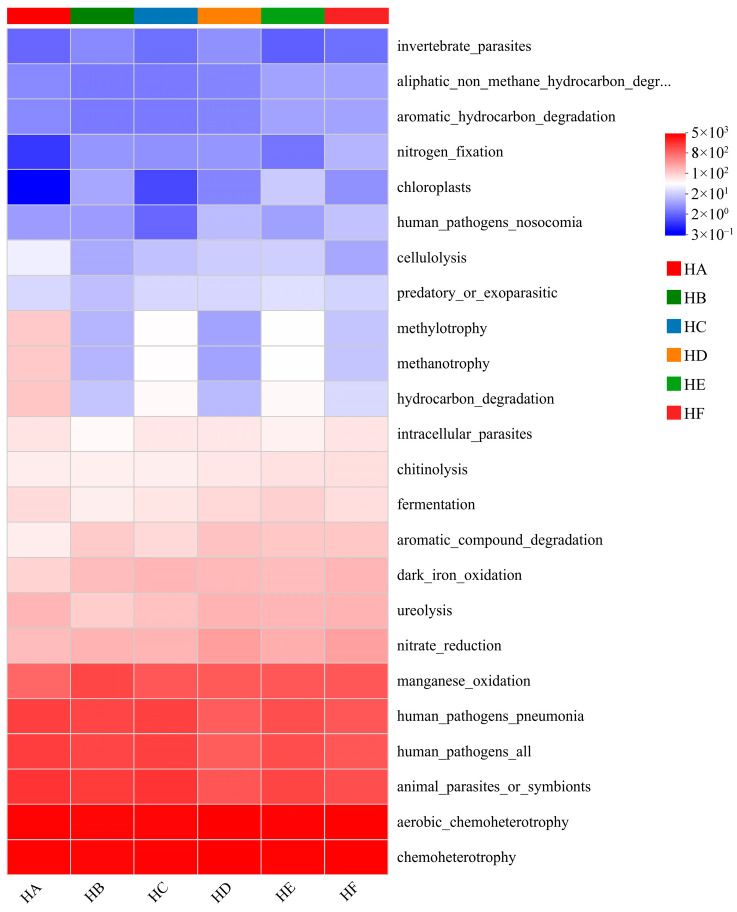
Heatmap visualization of FAPROTAX-predicted functional profiles across mixed-seeding treatments. Note: HA (monoculture of *E. nutans*), HB (binary mixture of *E. nutans* + *P. crymophila*), HC (ternary mixture of *E. nutans* + *P. crymophila* + *F. sinensis*), HD (four-species mixture of *E. nutans* + *P. crymophila* + *F. sinensis* + *P. poophagorum*), HE (five-species mixture of *E. nutans* + *P. crymophila* + *F. sinensis* + *P. poophagorum* + *F. kryloviana*), and HF (six-species mixture of *E. nutans* + *P. crymophila* + *F. sinensis* + *P. poophagorum* + *F. kryloviana* + *E. breviaristatus*).

**Figure 10 microorganisms-13-02341-f010:**
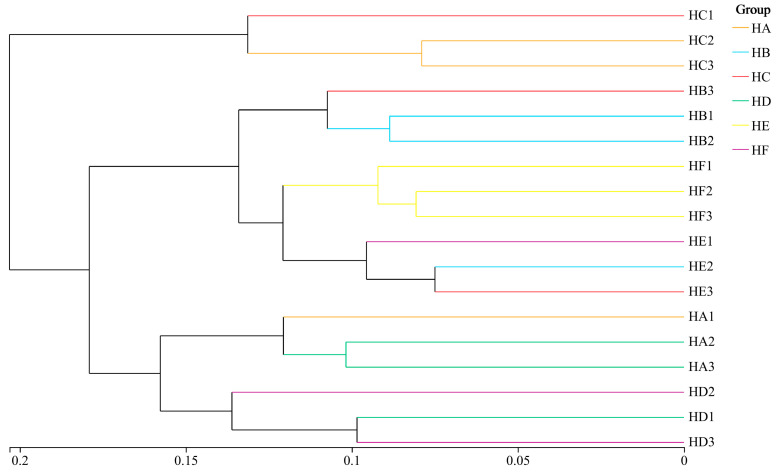
Hierarchical clustering of soil bacterial communities across mixed-seeding treatments. Note: HA (monoculture of *E. nutans*), HB (binary mixture of *E. nutans* + *P. crymophila*), HC (ternary mixture of *E. nutans* + *P. crymophila* + *F. sinensis*), HD (four-species mixture of *E. nutans* + *P. crymophila* + *F. sinensis* + *P. poophagorum*), HE (five-species mixture of *E. nutans* + *P. crymophila* + *F. sinensis* + *P. poophagorum* + *F. kryloviana*), and HF (six-species mixture of *E. nutans* + *P. crymophila* + *F. sinensis* + *P. poophagorum* + *F. kryloviana* + *E. breviaristatus*).

**Figure 11 microorganisms-13-02341-f011:**
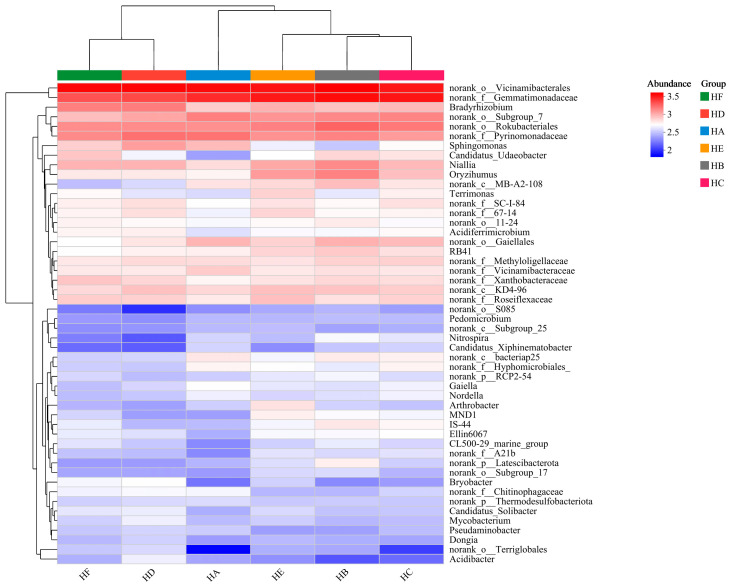
Heatmap visualization of soil microbial community composition across mixed-seeding treatments. Note: HA (monoculture of *E. nutans*), HB (binary mixture of *E. nutans* + *P. crymophila*), HC (ternary mixture of *E. nutans* + *P. crymophila* + *F. sinensis*), HD (four-species mixture of *E. nutans* + *P. crymophila* + *F. sinensis* + *P. poophagorum*), HE (five-species mixture of *E. nutans* + *P. crymophila* + *F. sinensis* + *P. poophagorum* + *F. kryloviana*), and HF (six-species mixture of *E. nutans* + *P. crymophila* + *F. sinensis* + *P. poophagorum* + *F. kryloviana* + *E. breviaristatus*).

**Figure 12 microorganisms-13-02341-f012:**
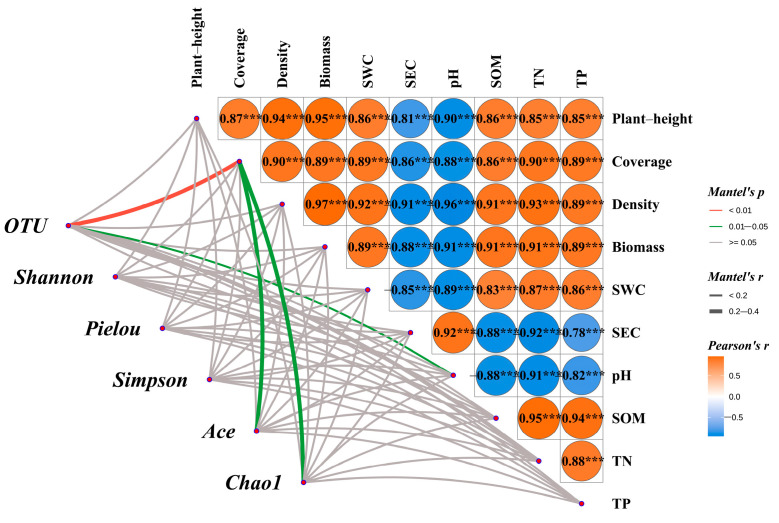
Mantel test analysis of vegetation characteristics, soil physicochemical properties, and soil bacterial communities. Note: SWC: soil water content; SEC: soil electrical conductivity; SOM: soil organic matter; TN: total nitrogen; TP: total phosphorus. The asterisks in the correlation heatmap denote statistical significance level as follows: *** *p* < 0.001.

**Figure 13 microorganisms-13-02341-f013:**
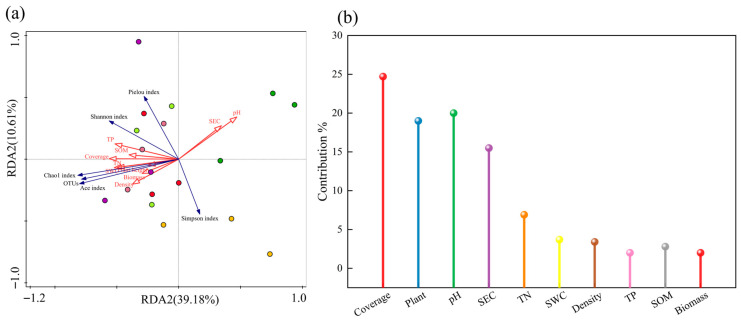
RDA triplot illustrating relationships between soil properties, vegetation characteristics, and bacterial community structure. Note: (**a**) Redundancy analysis (RDA) plot; (**b**) Contribution percentages of different factors. SWC: soil water content; SEC: soil electrical conductivity; SOM: soil organic matter; TN: total nitrogen; TP: total phosphorus.

**Figure 14 microorganisms-13-02341-f014:**
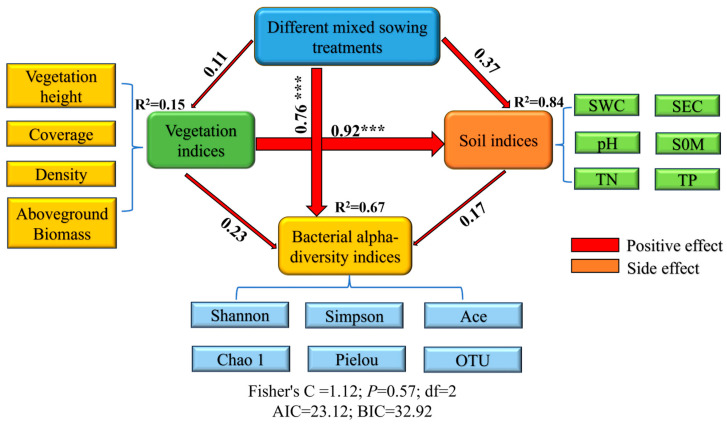
Structural equation model depicting treatment effects on vegetation characteristics, soil properties, and bacterial community structure. Note: Solid and dashed lines represent significant (*p* < 0.05) and non-significant pathways, respectively, with red/orange indicating positive/negative correlations. Line width corresponds to standardized path coefficients (β). SWC: soil water content; SEC: soil electrical conductivity; SOM: soil organic matter; TN: total nitrogen; TP: total phosphorus. *** denotes significance at *p* < 0.001 level.

**Figure 15 microorganisms-13-02341-f015:**
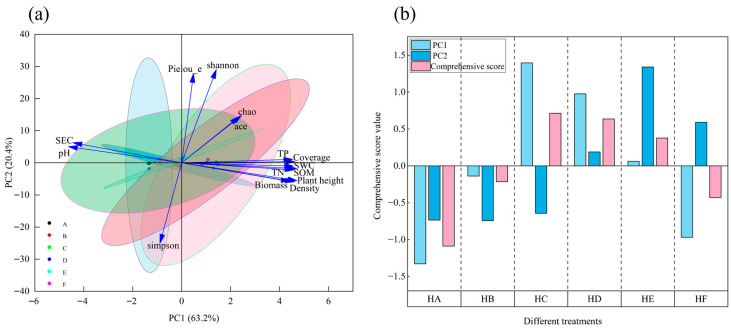
Multivariate evaluation of soil–plant–microbe systems. Note: (**a**) PCA biplot, (**b**) treatment score ranking. SWC: soil water content; SEC: soil electrical conductivity; SOM: soil organic matter; TN: total nitrogen; TP: total phosphorus.

**Table 1 microorganisms-13-02341-t001:** Experimental design of grass species mixtures for ecological restoration of degraded alpine grasslands.

No.	Treatment Code	Grass Species Combination	Mixed-Seeding Ratio
1	HA	*Elymus nutans* Griseb.	1
2	HB	*Elymus nutans* Griseb. + *Poa crymophila* Keng ex L. Liu cv. ‘Qinghai’	1:1
3	HC	*Elymus nutans* Griseb. + *Poa crymophila* Keng ex L. Liu cv. ‘Qinghai’ + *Festuca sinensis* Keng ex S. L. Lu cv. ‘Qinghai’	1:1:1
4	HD	*Elymus nutans* Griseb. + *Poa crymophila* Keng ex L. Liu cv. ‘Qinghai’ + *Festuca sinensis* Keng ex S. L. Lu cv. ‘Qinghai’ + *Poa poophagorum* Bor.	1:1:1:1
5	HE	*Elymus nutans* Griseb. + *Poa crymophila* Keng ex L. Liu cv. ‘Qinghai’ + *Festuca sinensis* Keng ex S. L. Lu cv. ‘Qinghai’ + *Poa poophagorum* Bor. + *Festuca kryloviana* Reverd. cv. ‘Huanhu’	1:1:1:1:1
6	HF	*Elymus nutans* Griseb. + *Poa crymophila* Keng ex L. Liu cv. ‘Qinghai’ + *Festuca sinensis* Keng ex S. L. Lu cv. ‘Qinghai’ + *Poa poophagorum* Bor. + *Festuca kryloviana* Reverd. cv. ‘Huanhu’ + *Elymus breviaristatus* Linn.	1:1:1:1:1:1

Note: HA (monoculture of *E. nutans*), HB (binary mixture of *E. nutans* + *P. crymophila*), HC (ternary mixture of *E. nutans* + *P. crymophila* + *F. sinensis*), HD (four-species mixture of *E. nutans* + *P. crymophila* + *F. sinensis* + *P. poophagorum*), HE (five-species mixture of *E. nutans* + *P. crymophila* + *F. sinensis* + *P. poophagorum* + *F. kryloviana*), and HF (six-species mixture of *E. nutans* + *P. crymophila* + *F. sinensis* + *P. poophagorum* + *F. kryloviana* + *E. breviaristatus*).

**Table 2 microorganisms-13-02341-t002:** Effects of different grass mixture treatments on vegetation height, coverage, density, and aboveground biomass in alpine grassland.

Year	Treatment	Vegetation Height/cm	Plant Coverage/%	Density/Plant·m^−2^	Aboveground Biomass/(g·m^−2^)
2023	HA	35.8 ± 1.17 ^c^	72.0 ± 2.65 ^d^	708 ± 18.7 ^d^	951 ± 19.5 ^c^
	HB	40.5 ± 1.40 ^b^	80.0 ± 4.36 ^bc^	791 ± 11.8 ^c^	1241 ± 77.1 ^b^
	HC	42.1 ± 0.97 ^b^	90.3 ± 3.51 ^a^	913 ± 25.5 ^a^	1580 ± 141.3 ^a^
	HD	45.9 ± 2.15 ^a^	92.3 ± 3.21 ^a^	855 ± 16.4 ^b^	1574 ± 94.8 ^a^
	HE	39.2 ± 0.83 ^b^	84.0 ± 3.61 ^b^	734 ± 21.7 ^d^	1154 ± 97.4 ^bc^
	HF	32.5 ± 2.21 ^d^	76.3 ± 3.21 ^cd^	651 ± 18.8 ^e^	978 ± 98.0 ^c^
2024	HA	32.0 ± 2.65 ^c^	75.3 ± 3.79 ^d^	665 ± 20.6 ^d^	1079 ± 131.2 ^c^
	HB	39.3 ± 2.08 ^b^	86.3 ± 3.21 ^bc^	800 ± 15.4 ^c^	1388 ± 95.4 ^b^
	HC	50.0 ± 2.65 ^a^	95.3 ± 2.08 ^a^	942 ± 17.9 ^a^	1645 ± 76.9 ^a^
	HD	45.0 ± 4.58 ^a^	90.7 ± 3.21 ^ab^	902 ± 18.5 ^b^	1431 ± 64.4 ^b^
	HE	34.0 ± 3.61 ^bc^	83.0 ± 4.36 ^c^	811 ± 20.5 ^c^	1206 ± 91.9 ^c^
	HF	28.3 ± 2.08 ^c^	77.0 ± 2.00 ^d^	692 ± 15.5 ^d^	1022 ± 124.2 ^c^

Note: HA (monoculture of *E. nutans*), HB (binary mixture of *E. nutans* + *P. crymophila*), HC (ternary mixture of *E. nutans* + *P. crymophila* + *F. sinensis*), HD (four-species mixture of *E. nutans* + *P. crymophila* + *F. sinensis* + *P. poophagorum*), HE (five-species mixture of *E. nutans* + *P. crymophila* + *F. sinensis* + *P. poophagorum* + *F. kryloviana*), and HF (six-species mixture of *E. nutans* + *P. crymophila* + *F. sinensis* + *P. poophagorum* + *F. kryloviana* + *E. breviaristatus*). Data are presented as mean ± standard deviation (SD) from three independent replicates. Different lowercase letters (e.g., “a”, “b”) indicate statistically significant differences among treatments at *p* < 0.05 based on one-way ANOVA with Tukey’s HSD post hoc test.

**Table 3 microorganisms-13-02341-t003:** Effects of grass species mixtures on soil physicochemical characteristics in alpine grasslands.

Year	Treatment	SWC/%	SEC/(μs·cm^−1^)	pH	SOM/(g·kg^−1^)	TN/(g·kg^−1^)	TP/(g·kg^−1^)
2023	HA	20.6 ± 0.77 ^d^	838 ± 37.5 ^a^	8.35 ± 0.11 ^a^	129 ± 9.87 ^b^	4.49 ± 0.14 ^e^	1.15 ± 0.16 ^c^
	HB	24.2 ± 0.64 ^c^	492 ± 38.6 ^cd^	7.70 ± 0.12 ^c^	139 ± 12.29 ^b^	4.45 ± 0.10 ^e^	1.23 ± 0.15 ^c^
	HC	27.5 ± 1.20 ^b^	428 ± 48.3 ^d^	7.34 ± 0.07 ^d^	197 ± 13.08 ^a^	7.16 ± 0.15 ^a^	1.84 ± 0.12 ^a^
	HD	30.5 ± 1.49 ^a^	564 ± 68.3 ^bc^	7.55 ± 0.19 ^cd^	187 ± 14.00 ^a^	6.42 ± 0.12 ^b^	2.01 ± 0.12 ^a^
	HE	25.0 ± 1.47 ^c^	618 ± 65.6 ^b^	8.02 ± 0.17 ^b^	152 ± 9.07 ^b^	6.16 ± 0.18 ^c^	1.51 ± 0.11 ^b^
	HF	22.5 ± 1.37 ^cd^	828 ± 38.4 ^a^	8.25 ± 0.13 ^ab^	129 ± 14.00 ^b^	5.01 ± 0.12 ^d^	1.22 ± 0.10 ^c^
2024	HA	20.8 ± 2.37 ^c^	832 ± 65.6 ^a^	8.22 ± 0.13 ^a^	132 ± 12.66 ^b^	4.38 ± 0.09 ^e^	1.28 ± 0.13 ^c^
	HB	24.6 ± 1.11 ^bc^	768 ± 43.5 ^ab^	7.81 ± 0.12 ^b^	147 ± 10.02 ^b^	6.13 ± 0.14 ^b^	1.53 ± 0.08 ^bc^
	HC	30.0 ± 2.23 ^a^	402 ± 63.3 ^c^	7.38 ± 0.12 ^d^	195 ± 17.79 ^a^	7.41 ± 0.15 ^a^	2.36 ± 0.17 ^a^
	HD	26.9 ± 1.35 ^ab^	649 ± 37.8 ^b^	7.53 ± 0.10 ^cd^	164 ± 13.01 ^b^	5.81 ± 0.12 ^c^	2.31 ± 0.20 ^a^
	HE	23.8 ± 1.64 ^bc^	709 ± 58.4 ^ab^	7.70 ± 0.11 ^bc^	153 ± 9.17 ^b^	5.10 ± 0.14 ^d^	1.65 ± 0.09 ^b^
	HF	20.3 ± 2.58 ^c^	802 ± 63.6 ^a^	8.13 ± 0.10 ^a^	141 ± 15.0 ^b^	4.54 ± 0.13 ^e^	1.42 ± 0.13 ^bc^

Note: HA (monoculture of *E. nutans*), HB (binary mixture of *E. nutans* + *P. crymophila*), HC (ternary mixture of *E. nutans* + *P. crymophila* + *F. sinensis*), HD (four-species mixture of *E. nutans* + *P. crymophila* + *F. sinensis* + *P. poophagorum*), HE (five-species mixture of *E. nutans* + *P. crymophila* + *F. sinensis* + *P. poophagorum* + *F. kryloviana*), and HF (six-species mixture of *E. nutans* + *P. crymophila* + *F. sinensis* + *P. poophagorum* + *F. kryloviana* + *E. breviaristatus*). Data are presented as mean ± standard deviation (SD) from three independent replicates. Different lowercase letters (e.g., “a”, “b”) indicate statistically significant differences among treatments at *p* < 0.05 based on one-way ANOVA with Tukey’s HSD post hoc test. SWC: soil water content; SEC: soil electrical conductivity; SOM: soil organic matter; TN: total nitrogen; TP: total phosphorus.

## Data Availability

The original contributions presented in this study are included in this article; further inquiries can be directed to the corresponding author.

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
