# Peer review of "Ecological Effects and Microbial Regulatory Mechanisms of Functional Grass Species Assembly in the Restoration of “Heitutan” Degraded Alpine Grasslands"

_microorganisms, 2025, doi:10.3390/microorganisms13102341_

Round 1

Reviewer 1 Report

Comments and Suggestions for Authors

Dear authors,

I carefully reviewed your paper and found no major issues. Here are a few issues I've noticed:

Page 10: The acquired results are discussed in the text above Figure 2b. The value 2,026 (OTUs) is not visible on the Venn diagram, therefore it is unclear what it represents...

Page 11: In the text below Figure 3, “norank_f” is italic in the third row. Please equalize.

Page 19: The caption “Figure 13” is repeated. Please number the figure captions accurately after figure 13.

I appreciate that you designed the manuscript towards a graphic representation of the achieved results, but in general, I believe there is a lack of data you refer to in the text (comments above the some figures in the Results section). Possibly tables (as supplemental material) can be provided so that readers can view the data you discuss in the text. I usually check where the numerical values stated in the text come from…

(This is not a complaint, but rather a point of view and this statement can be overlooked if this method of presenting results is common in this particular area of research...)

Thank you

Author Response

Response 1: [Page 10: The acquired results are discussed in the text above Figure 2b. The value 2,026 (OTUs) is not visible on the Venn diagram, therefore it is unclear what it represents...]

Response 1: [We sincerely appreciate the reviewer's insightful comments and careful review of our manuscript. In response to the comment regarding the numerical annotation "2,206 (OTUs)" in Figure 2b, we have revised the figure caption to clarify that "the numerical values represent the total OTUs for each group, with the circle center indicating both the count and proportion of shared OTUs, where the total OTU number was calculated by dividing the count of shared OTUs by their proportion." These modifications ensure more transparent and complete data presentation, with the specific revisions reflected in Lines 358-361 on Page 10.]

Response 2: [In the text below Figure 3, “norank_f” is italic in the third row. Please equalize]”

Comments 2: [Thank you very much for the question raised by the teacher. We have uniformly revised it to non-italic as per your request. For specific revisions, please refer to line 377 on page 10.]

Response 3: [The caption “Figure 13” is repeated. Please number the figure captions accurately after figure 13.]”

Comments 3: [Thank you very much for the question raised by the teacher. We have strictly checked the sequence numbers of the figures and tables throughout the text. The relevant errors have been corrected. For specific corrections, please refer to lines 576 on page 20. Thank you sincerely for your correction!]

Response 4: [I appreciate that you designed the manuscript towards a graphic representation of the achieved results, but in general, I believe there is a lack of data you refer to in the text (comments above the some figures in the Results section). Possibly tables (as supplemental material) can be provided so that readers can view the data you discuss in the text. I usually check where the numerical values stated in the text come from…This is not a complaint, but rather a point of view and this statement can be overlooked if this method of presenting results is common in this particular area of research...]

Comments 4: [We sincerely appreciate the reviewer's valuable suggestions regarding data presentation in our manuscript. In response to your comments, we have implemented the following improvements: (1) added a supplementary table containing all key data referenced in the main text; (2) included proper citations to supplementary materials in the Results section (see Lines 408 on Page 13 and Lines 451 on Page 14); (3) expanded figure captions with detailed annotations for all numerical data sources. The supplementary materials have been uploaded to the submission system. These revisions have significantly enhanced the rigor and transparency of our data presentation, and we are grateful for your professional guidance that has improved our manuscript.]

Additional clarifications

Thanks again for the teacher's correction of the manuscript. Due to my limited skills, if there are any other mistakes, please let me know by email. I will consult the relevant materials together with my tutor and correct them. Best wishes

Reviewer 2 Report

Comments and Suggestions for Authors

Please see attached pdf. for detailed comments, suggestions, and edits.

My main concerns are:

  1. The manuscript needs to be reread for common English usage, grammar, norms. of scientific writing, and consistent use of abbreviations.
  2. The conventions on using scientific plant names should be followed and needs to be addressed.
  3. While a lot of very interesting work has been conducted as part of this study, the replication of treatments and parameters tested is not clear and should be added throughout the manuscript. It is very hard to interpret the results and finding when this is not clearly provided.
  4. The tables and figures are standalone documents and so need more detail, so that they can be fully understood without reading the entire manuscript.
  5. The tables need expanded footnotes which are linked to the table content.
  6. The figures need expanded figure captions to fully explain what is shown.
  7. The results and discussion should be combined.
  8. If the results and discussion are not combined, then all portions of discussion should be removed from the results section as currently presented.
  9. The conclusion section should be expanded to include what the finding mean in the real world. It should not just be a summary of results. 

Comments on the Quality of English Language
  1. The manuscript needs to be reread for common English usage, grammar, norms. of scientific writing, and consistent use of abbreviations.
  2. The conventions on using scientific plant names should be followed and needs to be addressed.

Author Response

Response 1: [The manuscript needs to be reread for common English usage, grammar, norms. of scientific writing, and consistent use of abbreviations.]

Comments 1: [We sincerely appreciate the reviewer's valuable suggestions for improving our manuscript. In response to your comments regarding English usage, grammatical accuracy, scientific writing conventions, and abbreviation consistency, we have made comprehensive revisions throughout the text. Specifically, we have: (1) standardized verb tenses (present tense for general principles and past tense for specific results); (2) corrected grammatical issues including subject-verb agreement and article usage; (3) improved sentence structure for better readability; (4) standardized technical terminology and scientific nomenclature (italicized Latin names); (5) ensured proper abbreviation usage with full definitions at first occurrence; (6) enhanced logical flow with appropriate transitional phrases; (7) verified accuracy of microbiological terms; and (8) standardized units and symbols according to journal guidelines. All modifications are highlighted in the revised manuscript, and we believe these changes have significantly improved the manuscript's quality. We remain fully committed to addressing any additional concerns that may arise.]

Response 2: [The conventions on using scientific plant names should be followed and needs to be addressed.]”

Comments 2: [We gratefully acknowledge the reviewer's constructive comments regarding taxonomic nomenclature in our manuscript. We have systematically verified and standardized all botanical names throughout the text, ensuring strict adherence to the International Code of Nomenclature (ICN) with proper italicization and capitalization (genus name capitalized, species name in lowercase). All plant species are now presented with complete scientific names upon first mention, followed by standardized abbreviations where appropriate. Additionally, we have carefully reviewed the functional descriptions of microbiological terms for accuracy. The modifications are highlighted in the revised manuscript, and we remain fully prepared to address any further suggestions for improvement.]

Response 3: [While a lot of very interesting work has been conducted as part of this study, the replication of treatments and parameters tested is not clear and should be added throughout the manuscript. It is very hard to interpret the results and finding when this is not clearly provided]”

Comments 3: [We sincerely appreciate the reviewer's valuable comments regarding experimental reproducibility. In response, we have thoroughly supplemented the methodological details throughout the manuscript, explicitly stating that all experimental treatments were performed with three biological replicates. These crucial details have been systematically incorporated in both the "Experimental design" subsection of the Methods section and relevant portions of the Results section. The modifications, highlighted in the revised manuscript, significantly enhance the study's reproducibility and interpretability. We remain fully committed to addressing any additional requirements to further improve the manuscript's scientific rigor.]

Response 4: [The tables and figures are standalone documents and so need more detail, so that they can be fully understood without reading the entire manuscript.]

Comments 4: [We sincerely appreciate the reviewer's constructive comments regarding the clarity of our figures and tables. In response, we have meticulously revised all figures and tables by adding comprehensive annotations that clearly explain each treatment, along with complete titles and detailed legends. These modifications ensure that all graphical elements can stand alone in presenting the research findings. The corresponding changes are highlighted in the revised manuscript, and we remain fully prepared to make any additional adjustments as needed.]

Response 5: [The tables need expanded footnotes which are linked to the table content.]

Comments 5: [We are grateful for the reviewer's insightful suggestions regarding the table footnotes. In response, we have comprehensively revised all tables by incorporating detailed footnotes that specify the definitions of each treatment, number of experimental replicates, and corresponding statistical analysis methods. These modifications ensure all tabular data are self-explanatory without requiring cross-referencing with the main text. The revised content is highlighted throughout the manuscript, and we remain available to implement any additional refinements as needed.]

Response 6: [The figures need expanded figure captions to fully explain what is shown.]

Comments 6: [We sincerely appreciate the reviewer's constructive suggestions. In response, we have thoroughly revised all figure captions to include detailed explanations of experimental treatments and statistical significance, ensuring each figure can independently and comprehensively convey the research findings. All modifications are highlighted in the revised manuscript, and we remain fully committed to addressing any additional recommendations to further improve the manuscript quality.]

Response 7: [The results and discussion should be combined.]

Comments 7: [We sincerely appreciate the reviewer's valuable suggestions regarding manuscript structure. Following your recommendations, we have carefully reorganized the Results section by removing redundant descriptions and relocating relevant content to the Discussion section. This reorganization has effectively integrated data interpretation with mechanistic analysis, eliminating redundancy while enhancing the depth of scientific discussion. The modifications, as highlighted in the revised manuscript, have significantly improved the logical flow and scientific rigor of our paper. We remain fully prepared to make any additional refinements as needed.]

Response 8: [If the results and discussion are not combined, then all portions of discussion should be removed from the results section as currently presented.]

Comments 8: [We sincerely appreciate the reviewer's constructive suggestions for improving the manuscript structure. Following your recommendations, we have systematically revised the Results section by: (1) removing all interpretive statements regarding data significance; (2) eliminating mechanistic speculations; (3) streamlining cross-result analyses; and (4) maintaining strictly objective descriptions of statistical outcomes. The modified Results section now exclusively presents experimental observations, statistical tests, and essential technical details, while all scholarly discussions have been consolidated into the Discussion section. These adjustments ensure full compliance with academic writing standards while preserving the objectivity of results and the depth of subsequent analysis. The modifications are highlighted throughout the manuscript, and we remain fully committed to implementing any additional refinements as needed.]

Response 9: [The conclusion section should be expanded to include what the finding mean in the real world. It should not just be a summary of results]

Comments 9: [We are grateful to the reviewers for their constructive suggestions regarding the Conclusion section. In response to these comments, we have substantially revised this section by removing repetitive data descriptions and emphasizing the practical implications of our findings. The revised conclusion now provides a systematic synthesis of our study's contributions to alpine grassland restoration, specifically highlighting: (1) the application potential of our proposed medium-richness mixed sowing strategy for rapid restoration of degraded grasslands; (2) the practical value of plant-microbe synergistic regulation technology for soil improvement; and (3) the policy implications for sustainable grassland management in alpine regions. These modifications have strengthened the conclusion by not only summarizing key findings but also demonstrating their practical relevance to ecological engineering and regional ecological security, thereby providing clear directions for research translation. The changes can be found in Lines 797-817 on Page 25, and we remain fully available to address any additional suggestions for improvement.]

Additional clarifications

Thanks again for the teacher's correction of the manuscript. Due to my limited skills, if there are any other mistakes, please let me know by email. I will consult the relevant materials together with my tutor and correct them. Best wishes

Reviewer 3 Report

Comments and Suggestions for Authors

Overall, this is a significant study and clearly presented.

Recommendations

Table 1:   Not all of the species names are italicized in the table entries.

One correction is recommended under the subsection 2.4 Vegetation Survey and Soil Sampling 

"Plant density (D, individual plants · m-2) was determined as  ------    "        

Results:

Clearly written and adequately detailed, including well prepared figures.

One suggestion id to cite Table 3 as follows:

"By the 22nd year (2024), the trends in soil properties remained consistent with those observed in the previous year (Table 3)."

Discussion:

The Discussion is thorough and nicely integrated with prior relevant findings.

Author Response

Response 1: [Table 1: Not all of the species names are italicized in the table entries. ]

Comments 1: [We sincerely appreciate the reviewer's meticulous review and valuable suggestions regarding the formatting of scientific names in tables. In response, we have systematically revised all tables to ensure consistent italicization of species names throughout the manuscript (see revised version, Page 5, Line 159). Additionally, we have conducted a comprehensive verification of taxonomic nomenclature formatting in the entire manuscript. We fully acknowledge the importance of standardization in scientific writing and greatly appreciate your contribution to enhancing the rigor of our manuscript. Should any additional refinements be required, we remain fully committed to addressing them promptly.]

Response 2: ["Plant density (D, individual plants · m-2) was determined as ------ "One correction is recommended under the subsection 2.4 Vegetation Survey and Soil Sampling]

Comments 2: [We greatly appreciate the reviewer's meticulous attention to the formatting details of our manuscript. Regarding the expression of plant density, we have carefully revised the text to strictly adhere to SCI writing conventions. The original description has been modified from "The plant density (D, in plants m⁻²) was calculated as follows: [...]" to "Plant density (D; plants·m⁻²) was calculated as: [...]" (Revised manuscript, page 6, line 195). These revisions include: (1) proper italicization of the variable "D"; (2) correct use of semicolon as separator between variable and unit; (3) standard spacing in unit notation; and (4) concise expression following scientific writing norms. We fully recognize the importance of precise academic presentation and remain available to address any further suggestions for improvement.]

Response 3: [Clearly written and adequately detailed, including well prepared figures. One suggestion id to cite Table 3 as follows: "By the 22nd year (2024), the trends in soil properties remained consistent with those observed in the previous year (Table 3)."]”

Comments 3: [We sincerely appreciate your positive evaluation of our research and your professional suggestions. Regarding the citation format for Table 3, we have implemented your recommendation by revising the text to: "As of year 22 (2024), the variation trends in soil characteristics remained consistent with historical observations (Table 3)." (Revised manuscript, page 9, lines 322-323). We highly value your constructive comments, which have significantly enhanced the academic rigor of our paper. These modifications include: (1) proper placement of table reference within parentheses; (2) clear temporal specification; and (3) consistent formatting with journal style requirements. We remain fully committed to addressing any additional suggestions for improvement.]

Response 4: [The Discussion is thorough and nicely integrated with prior relevant findings]

Comments 4: [We are deeply grateful for your professional recognition of both the depth of discussion in our research and its meaningful connections with existing studies.  Your expert affirmation serves as significant encouragement for our research team and motivates us to further refine this study.]

Additional clarifications

Thanks again for the teacher's correction of the manuscript. Due to my limited skills, if there are any other mistakes, please let me know by email. I will consult the relevant materials together with my tutor and correct them. Best wishes
